# Organ-Specific Endothelial Cell Differentiation and Impact of Microenvironmental Cues on Endothelial Heterogeneity

**DOI:** 10.3390/ijms23031477

**Published:** 2022-01-27

**Authors:** Laia Gifre-Renom, Margo Daems, Aernout Luttun, Elizabeth A. V. Jones

**Affiliations:** 1Centre for Molecular and Vascular Biology, Department of Cardiovascular Sciences, Katholieke Universiteit Leuven (KU Leuven), BE-3000 Leuven, Belgium; laia.gifrerenom@kuleuven.be (L.G.-R.); margo.daems@kuleuven.be (M.D.); aernout.luttun@kuleuven.be (A.L.); 2Department of Cardiology, CARIM School for Cardiovascular Diseases, Maastricht University, 6229 ER Maastricht, The Netherlands

**Keywords:** endothelial cell, vascular development, heterogeneity, organ-specific signature, phenotypic drift, microenvironment, mechanobiology, extracellular matrix

## Abstract

Endothelial cells throughout the body are heterogeneous, and this is tightly linked to the specific functions of organs and tissues. Heterogeneity is already determined from development onwards and ranges from arterial/venous specification to microvascular fate determination in organ-specific differentiation. Acknowledging the different phenotypes of endothelial cells and the implications of this diversity is key for the development of more specialized tissue engineering and vascular repair approaches. However, although novel technologies in transcriptomics and proteomics are facilitating the unraveling of vascular bed-specific endothelial cell signatures, still much research is based on the use of insufficiently specialized endothelial cells. Endothelial cells are not only heterogeneous, but their specialized phenotypes are also dynamic and adapt to changes in their microenvironment. During the last decades, strong collaborations between molecular biology, mechanobiology, and computational disciplines have led to a better understanding of how endothelial cells are modulated by their mechanical and biochemical contexts. Yet, because of the use of insufficiently specialized endothelial cells, there is still a huge lack of knowledge in how tissue-specific biomechanical factors determine organ-specific phenotypes. With this review, we want to put the focus on how organ-specific endothelial cell signatures are determined from development onwards and conditioned by their microenvironments during adulthood. We discuss the latest research performed on endothelial cells, pointing out the important implications of mimicking tissue-specific biomechanical cues in culture.

## 1. Introduction

Vascular endothelial cells line the entire circulatory system and show remarkable heterogeneity. Even though endothelial cells originate from the same progenitor cells during development, they eventually contribute to different subtypes of endothelia. Based on morphology, the microvasculature consists of three main phenotypes: discontinuous, fenestrated, and non-fenestrated endothelium (Figure 1). These morphological differences correlate with vascular permeability and contribute to organ-specific functions. Non-fenestrated endothelium has a low permeability, which is found in brain, heart, and lung microvessels, as well as within all larger vessels (i.e., arteries and veins). A fenestrated endothelium has transcellular pores of about 70 nm in diameter [1], which are covered by a thin, non-membranous diaphragm. As fenestrae are associated with increased filtration and trans-endothelial transport functions, they are found in kidney, endocrine glands, and gastric and intestinal mucosa. Lastly, discontinuous endothelium is found exclusively in sinusoidal endothelium, such as the bone marrow, the spleen, and the liver endothelium. The latter has larger fenestrations, up to 100 or 200 nm in diameter, that are devoid of a diaphragm and have large pores within individual cells [1]. Additionally, the underlying basement membrane is only partly developed, resulting in a high permeability. Each organ is made up of different types of endothelia. In the kidney, fenestrated endothelium in the peritubular capillaries and glomeruli ensures proper filtration, while continuous endothelium elsewhere provides the kidney itself with nutrients and oxygen [2]. Similarly, circumventricular organs of the brain are lined with fenestrated endothelium, while elsewhere, the tight blood–brain barrier (BBB) is found [3]. 

The endothelium is equipped with structural components involved in endocytosis, transcytosis, and proper cell–cell and cell–cytoskeleton contact. The presence of these structural components differs greatly between organ systems. While the BBB is extremely enriched with tight junctions, they gradually become looser when moving from large arteries towards the capillaries. However, in post-capillary venules, tight junctions are rather disorganized, facilitating the inflammation-induced extravasation of leukocytes [4]. Furthermore, the liver sinusoidal endothelium is equipped with high amounts of clathrin-coated pits and vesicles that aid in endocytosis [5]. Similarly, caveolae are most abundant in continuous, non-fenestrated capillaries, such as heart and lung tissue, while they rarely appear in the BBB, as they ensure proper transcytosis across the endothelium. These characteristics affect vascular permeability and heterogeneity across organs.

Endothelial heterogeneity is induced and maintained by the surrounding microenvironment, which is already present during embryonic development. In the adult organism, endothelial cells are influenced by both mechanical and biochemical cues derived from the tissue. Examples of these cues are the composition of the basement membrane, tissue stiffnesses, blood flow rates, pulsatility, and the close contact with their neighboring cells, such as cardiomyocytes in the heart or astroglial cells in the brain. However, these are all nonheritable changes that result in transcription factor-induced gene expression and posttranslational modifications. Endothelial heterogeneity is also regulated by epigenetic modifications, which are preserved during mitosis and in the absence of extracellular cues. Although endothelial cells quickly lose their organ-specific properties when cultured [6,7], several DNA microarray studies revealed site-specific signatures that remain after several passages, demonstrating the importance of epigenetics for endothelial heterogeneity [8,9]. 

There has been a great effort in understanding endothelial heterogeneity as researchers attempt to mimic these specific types of endothelia in vitro. Understanding how endothelial cells acquire and maintain their heterogeneity will allow us to move towards more specialized vascular research and an organ-oriented treatment strategy. In this review, we discuss how endothelial cell heterogeneity is determined during development and adulthood, as well as the recent advances in the study and characterization of organ-specific endothelial cell signatures. We also raise awareness on the importance of implementing microenvironmental cues in vascular research in order to obtain more relevant and organ-specific readouts towards more specialized tissue engineering approaches.

## 2. Development of Organ Specificity among Endothelial Cells

### 2.1. Vasculogenesis—Intrinsic Versus Extrinsic Factors

Endothelial cells have a mesodermal origin; during vasculogenesis, a “first draft” of the vascular system is laid down to support the growing embryo [10]. In vertebrates, vasculogenesis is initiated in the blood islands at the distal aspect of the yolk sac. The blood islands give rise to both primitive endothelial cells, called angioblasts, and primitive red blood cells, erythroblasts [11,12] (Figure 2). Extraembryonic angioblasts subsequently migrate over the yolk sac to form a randomly organized primitive vascular plexus. VEGF signaling is abundantly studied as a critical mediator of vasculogenesis [13,14,15,16]. Both heterozygous and homozygous *Vegf-A* null mice died during embryonic development, at E11.5 and E9.5, respectively, due to impaired angiogenesis and disrupted formation of the blood islands [13,14]. Similarly, *Vegfr2^−/−^* embryos die between E8.5 and E9.5 due to defects in vasculogenesis and the lack of blood islands [15]. Vasculogenesis is tightly regulated by a family of E26 transformation-specific (ETS) transcription factors [17]. 

Although a certain redundancy has been described, ETV2 drives both *Vegfr2* and *Tie2* expression in endothelial progenitor cells [18,19]. Already at E8.5, *Etv2^−/−^* embryos present with impaired vasculogenesis and reduced VEGFR2 expression, while they die between E9.0 and E10.5 [18,19]. Although VEGF signaling drives endothelial development, its own expression is regulated by the microenvironment of the developing tissue. In lung tissue, VEGF is initially secreted by mesenchymal cells, while later, the main source shifts to alveolar epithelium [20,21,22]. Between E9.5 and E12, a balance between FGF9 expressed by epithelial cells and FGF10 expressed by the surrounding mesenchymal cells sustains VEGF expression. FGF10 drives mTORC1/Sprouty2 signaling in epithelial cells, which initiates the production of VEGF by the epithelium [23]. In turn, FGF9 binds FGFR1 on epithelial cells and induces the expression of VEGF [24]. Similarly, in the developing heart, cardiomyocytes are an important source of VEGF for the coronary vessels; embryos with a cardiomyocyte-specific *Vegf* ablation present with fewer coronary vessels and a thinned ventricular wall at E13.5, suggesting that the tissue-specific microenvironment regulates endothelial differentiation by regulating VEGF expression [25]. Furthermore, endothelial differentiation is at least partly driven by FGF2 and BMP4, although they are classically needed to induce the mesoderm. Loss of BMP4, as well as downstream effectors SMAD5 and SMAD4, prevents the induction of mesoderm and as a result, these embryos lack an organized yolk sac vasculature [26,27,28,29]. Moreover, BMP4 alone is sufficient to induce endothelial formation in the absence of endodermal cues, suggesting that BMP4 drives mesodermal formation [30]. However, vasculogenesis alone does not ensure that the vasculature expands, remodels, and adapts according to the requirements of the developing embryo. During development and throughout adult life, new blood vessels are created from the existing vasculature during either sprouting angiogenesis or intussusceptive angiogenesis [31].

**Figure 2 ijms-23-01477-f002:**
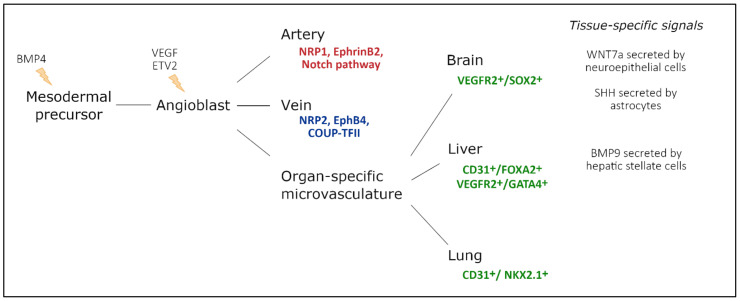
Endothelial cell heterogeneity throughout embryonic development and main factors involved. Mesodermal cells differentiate into vascular and hematopoietic progenitors under the influence of BMP4, secreted by the endoderm. These endothelial progenitor cells differentiate under the influence of VEGF-signaling, driven by ETV2 expression. In turn, they populate arteries, veins, and capillaries. During arteriovenous differentiation, endothelial cells take on an arterial or venous identity, characterized by expression of *Nrp1*, *EphrinB2*, and *Notch* signaling components, or *Nrp2*, *EphB4*, and *Coup*-*TFII* expression, respectively. Endothelial cells from capillaries are influenced by the microenvironment of their respective organ. Furthermore, a subset of organ-specific endothelial cells also has non-mesodermal origins, based on co-expression of endothelial markers and tissue-specific endoderm markers. Adapted from [32].

### 2.2. Large Conduit Vessel Differentiation

Although all endothelial cells originate from the same type of progenitor cell, there is a big discrepancy between large conduit vessels (arteries or veins) and the microvasculature. After the onset of blood flow, the primitive vascular plexus is remodeled into a hierarchical network, including arteries, veins, and capillaries. Although the heart tube already beats by E8.0, proper blood flow is only present once erythrocytes are released from the blood islands at E8.5, with vascular remodeling occurring over the following day [33,34]. In the absence of erythrocytes or blood flow altogether, the primitive plexus fails to remodel into differentiated arteries and veins [33,35]. Moreover, restoring blood viscosity by injecting hetastarch intravascularly in embryos lacking erythrocytes is sufficient to rescue vascular remodeling, highlighting the crucial role of mechanical forces in creating a proper vascular hierarchy [33]. 

Arterial and venous differentiation is the earliest demonstration of endothelial heterogeneity in the embryo [8]. It is present before the onset of flow [36]; however, shear stress can override this initial identity. Ligation of vitelline arteries in chicken embryos results in a venularization within 24 h, as evidenced by downregulation of *Gja5*, *Nrp1*, and *EphrinB2* and upregulation of *Coup*-*TFII*, *Nrp2*, and *Tie2* [35,37]. Strikingly, reperfusion completely restores the initial arterial gene expression, which highlights the plasticity of endothelial identity and the importance of shear stress for arteriovenous differentiation [35,37]. 

### 2.3. Organ-Specific Vascular Development

As organogenesis commences and tissues specialize into organs with specific morphologies and phenotypes, endothelial cells are no longer exclusively influenced by intrinsic factors. Along embryonic development, newly formed endothelial cells are exposed to extrinsic factors that induce a tissue-specific endothelial differentiation program (Figure 2). 

Within the developing brain, WNT7a/b secreted by the neuroepithelial cells binds endothelial WNT receptors Frizzled 4/6/8 and stabilizes β-catenin (encoded by *Ctnnb1*). Neuroepithelial-specific loss of WNT7a/b signaling induces severe hemorrhages within the central nervous system (CNS) exclusively, demonstrating that the neural microenvironment influences the surrounding vasculature [38]. Endothelial-specific *Ctnnb1^−/−^* and *Lrp5/6^−/−^* embryos present with a similar phenotype, while the neural tissue completely lacks endothelial cells when endothelial β-catenin is destabilized [39,40]. WNT signaling is thought to drive a specific CNS endothelial differentiation program, including the expression of typical membrane receptor and transporter GLUT1. Impaired WNT7a/b signaling directly reduces GLUT1 expression levels in neural endothelial cells, while ectopic WNT7a drives GLUT1 expression in endothelial cells outside of the CNS [38]. Moreover, in the absence of *Ctnnb1*, the expression of several typical transcription factors declines in neuronal endothelial cells, demonstrating a key role for β-catenin in activating a BBB-specific differentiation [41]. 

In contrast to WNT signaling, the effects of GPR124 signaling are confined to specific regions of the developing CNS [42,43]. GPR124 is thought to regulate endothelial cell sprouting and migration in the forebrain and spinal cord exclusively. *Gpr124^−/−^* embryos show reduced sprouting in the forebrain region and vascular patterning defects by E11.5 and die by E15.5 [42,43]. GLUT1 expression in the forebrain is reduced in *Gpr124^−/−^* embryos, demonstrating that the acquisition of BBB markers may be coupled to developmental brain angiogenesis [42]. Furthermore, sonic hedgehog (SHH) is secreted by perivascular astrocytes and regulates the expression of several tight junction proteins, including Occludin, VE-cadherin, Claudin3, and Claudin5 [44]. Similarly, endothelial-specific loss of Smoothened (*Smo*), a downstream target of SHH signaling, results in an increased BBB permeability at E14, accompanied by a decreased expression of Occludin, Claudin3, Claudin5, and ZO1 [44]. 

In the liver, GATA4 has been identified as an important regulator of liver sinusoidal endothelial cell (LSEC) specification [45]. Deleting *Gata4* in STAB2^+^ or LYVE1^+^ cells results in embryonic lethality between E15.5 and E17.5, while a severely hypoplastic liver is observed at E11.5. LSEC-specific *Gata4*^−/−^ embryos undergo a major switch from a sinusoidal phenotype to a continuous capillary phenotype, accompanied by the formation of a basement membrane [45]. Furthermore, endothelial VE-Cadherin is upregulated in LSEC-specific *Gata4*^−/−^ embryos, demonstrating an increased stability of adherens junctions in the liver microvasculature. Recently, BMP9, secreted by hepatic stellate cells that ensheath the liver sinusoids, was identified as one of the paracrine regulators of LSEC fenestration by regulating GATA4 expression in LSECs [46].

Although endothelial cells generally have a mesodermal origin, a subset seems to be derived from tissue-specific endodermal cells. Between E10.5 and E14, a subset of VEGFR2^+^/SOX2^+^ endothelial cells is observed within the developing brain, which is lost by E18.5 [47]. Similarly, at E9.5, some rare CD31^+^FOXA2^+^ cells are observed in the liver buds; however, by E12.5, they evolve into an evenly dispersed population of CD31^+^FOXA2^+^ cells [48]. Furthermore, a subset of GATA4^+^/VEGFR2^+^ double-positive cells has been identified between E10.5 and E14 in the budding liver, suggesting a common progenitor cell with hepatocytes [47]. Single-cell analysis of the developing liver predicted that also DLL4, VEGFA, and the TGF-β signaling pathway affect LSEC identity, regulating, amongst others, the expression of *Icam2*, *Sparc*, and *Lyve1* [49]. 

Within the lung, a subset of endothelial cells is thought to derive from NKX2.1^+^ endodermal cells [50]. Already at E10.5, a subset of endothelial cells in the lung co-expresses CD31 and NKX2.1, which is lost by E18.5 [47]. NKX2.1-specific *Vegfr2^-/-^* embryos present with a decreased endothelial population [47]. Thus, it seems that a subset of endothelial cells is derived from a similar progenitor type as their surrounding tissue, committing to the vascular lineage over the course of development. 

## 3. Technological Progress in Assessing Endothelial Cell Heterogeneity

Independent of which vascular bed or organ they reside in, each endothelial cell’s specific features are dictated by a unique panel of molecular determinants which translate into characteristic morphological hallmarks that accommodate the specific functions of the host organs [32,51,52]. Rather than giving an extensive overview of the current knowledge of specific molecular, morphological and functional features of endothelial cells in different organs (for which we refer to a selection of seminal and recent reviews) [32,51,52,53,54,55,56,57,58], we use the liver endothelium to demonstrate how rapid technological advances have revolutionized the field of endothelial cell heterogeneity and have caused an exponential rise in the number of reports on this topic in the literature (Figure 3). 

The earliest studies documenting heterogeneity among endothelial cells were from the 1950′s and mostly focused on particular morphological features that could be visualized at subcellular resolution by transmission electron microscopy [32,59]. The presence of non-diaphragmed fenestrae in LSECs, possibly the most prototypic hallmark of specialization in endothelial cells, was first described by Eddie Wisse in 1970 (Figure 3A) [60]. Another elegant technique to depict the organ-specific structural organization of (micro)vascular endothelial cell networks in different organs is vascular casting combined with scanning electron microscopy or micro-computed tomography. This combination shows how the sinusoidal vessels bridge several portal venules with a central venule within the liver lobule, the functional unit of the liver (Figure 3A) [61,62,63]. The same technology has been recently used to reveal that expansion of the sinusoidal network in the liver can occur through a less common blood vessel formation modus, i.e., intussusceptive angiogenesis [64]. Specific posttranslational protein modifications, such as glycosylation, can also be used to identify heterogeneity, such as by differential binding of natural glycan ligands, i.e., lectins (Figure 3B) [65]. Together with the larger size and lower frequency of fenestrae, the preferential ability of periportal LSECs to bind certain lectins represented the first evidence for zonal differences in LSEC characteristics [65,66,67]. 

Later on, the sequencing of the entire genome of several species, including humans, and the development of protocols to isolate endothelial cells from different tissues enabled genome-wide expression profiling at the RNA level of these cells. There has been an impressive evolution in this area on different levels. First, the earliest studies establishing differential genome-wide expression profiles of organ-derived endothelial cells used a bulk approach. This consisted in profiling by sequential analysis of gene expression (SAGE) [66], microarray [58,68,69,70] or (later) by RNA sequencing (recently archived in EndoDB, an elaborate database of endothelial transcriptomics) [71] on all endothelial cells of an organ isolated by ‘pan-endothelial’ markers (e.g., PECAM1, VE-cadherin or Tie2; Figure 3B). Two important caveats, however, are that the literature is not always unequivocal in terms of the expression of certain markers [59,72,73], and that species-specific differences have been reported, e.g., in the case of plasmalemmal vesicle-1 (PV-1 or PLVAP), which is present on adult mouse but not human LSECs [64,73,74], or in the different expression levels of selenoprotein between human and pig HUVECs [75]. Similarly, functional differences between mouse and human lung microvascular endothelial cells have been reported, including differences in microfilament alignment and barrier permeability in response to inflammatory conditions [76]. As new therapeutics are traditionally developed in animal models prior to clinical testing, these species-specific differences must be considered during their translation to patients.

More recently, by changing the resolution of transcriptomic profiling to the single-cell level (scRNAseq; recently reviewed for the vascular system in general in ref. [53] and the liver in particular in ref. [77]), it has become clear that endothelial cell heterogeneity does not stop at the boundaries of different organs but that there is a significant level of intra-organ heterogeneity [78,79]. In the case of the liver, the earlier mentioned zonal differences seen in the sinusoidal endothelium by morphological and histological analyses were confirmed at the transcriptional level by scRNAseq (Figure 3B) [74,80,81,82,83]. Importantly, since scRNAseq can simultaneously document the expression profile of all cells in an organ, the reciprocal communication between endothelial and non- endothelial cells that co-determines their specific characteristics can also be analyzed [55,74,83,84]. Furthermore, this technology also allows for the detection of alterations in the cellular landscape, where certain endothelial cell subtypes may expand, newly arise, or disappear only upon pathological challenges [74,85,86]. For instance, it has been shown that the liver lymphatic endothelial cell population significantly expands after a fibrotic challenge [85,87]. 

One important drawback of scRNAseq, caused by the necessity to (enzymatically) dissociate the tissue of interest, is the potential differential sensitivity of cell types (including endothelial cell types) to the dissociation protocol, causing loss of these cell types in the analysis [88,89,90]. Single-nuclei RNA sequencing (snRNAseq) can overcome this dissociation bias as the nuclear membrane seems more resistant to tissue dissociation than the cell membrane. Furthermore, since nuclei, unlike cells, also are resistant to freeze-thaw manipulation, snRNAseq can also be performed on frozen archived material. While multiple studies using snRNAseq have been reported on the heart [91], only two studies using snRNAseq on the liver have been reported to our knowledge [92,93]. 

Another gap in scRNAseq is that spatial information is lost. Spatial transcriptomics has been recently developed to overcome this problem, although the resolution of this technology may not (yet) be sufficient to allow for analysis at a single-cell level [94]. In a recent study, Hou et al. have integrated spatial transcriptomics with scRNAseq to reveal the in situ expression profile of the developing human liver [95]. Alternatively, in the liver, spatial information related to LSEC zonation has been delivered by paired-cell sequencing, thereby using the zonated hepatocyte expression pattern as a landmark or by spatial sorting based on markers previously shown to have a zonated endothelial expression pattern, such as cKit (Figure 3B) [81,96]. 

A second revolution in the ‘omics area’ is the departure from a ‘transcriptomics-centric’ view through combining/cumulating different ‘omics’ analyses, such that information is obtained not only on the transcriptome, but also on the proteome and the epigenome (Figure 3B) [92,93,96,97,98,99,100,101,102,103]. At this time, however, single-cell analyses at the level of proteins and their post-translationally modified versions remain a technical challenge. As referred to above, in a recent study, to overcome this problem, Inverso et al. have designed an unprecedented method based on spatial sorting in order to enable zonated multiomics analyses of liver endothelial subclusters defined by scRNAseq [96]. Using this method, they created a multiomic map of the liver lobular vasculature (which is available as an interactive web tool) and identified a zonated protein activation (phosphorylation) pattern, including for tyrosine kinase Tie1, which regulates zonation through Wnt-signaling. While proteomic analyses are still beyond single-cell resolution [104], analysis of the epigenetic status at single-cell resolution is already possible by single-cell assay for transposase-accessible chromatin using sequencing (scATACseq; Figure 3B) [99,102,103]. Integrating transcriptomic data (scRNAseq) with information on chromatin accessibility (scATACseq) allows for the unraveling of gene regulatory networks (‘regulomes’) governing cellular heterogeneity [105]. In the liver, this integration has been recently applied to document functional heterogeneity among hepatocytes, but non-parenchymal cells (like endothelial cells) were not included in this study [102].

In addition to morphology and molecular expression, specialized functions are also part of the organ-specific endothelial cell signature [32,57,106]. In many cases this functional information can be derived by functional annotation analysis of the transcriptomic fingerprint [68,69,78,80,82,87,102,106,107,108]. The identified functions can then be tested. As for LSECs, (specific) functions that can be tested include (Figure 3C): (i) the binding, uptake and lysosomal degradation of fluorescently labeled macromolecules (reflecting the presence of scavenger receptors such as mannose receptor or MRC1, the Fc gamma receptor IIb or Stabilins) [59,107,109,110,111,112], (ii) the binding of viral proteins (reflecting the expression of receptors interacting with these viruses such as L-SIGN) [107]; and (iii) the production of coagulation Factor VIII [110,113]. 

Alternatively, an organ-specific functioning of endothelial cells can be tested by their ability to support the function of the organ’s parenchymal cells in co-culture models, e.g., albumin production by hepatocytes in the liver [106]. A recent study has shown that the artificial overexpression of LSEC-overrepresented transcription factors in non-specialized endothelial cells is not sufficient for a full restoration of the functional, morphological, and molecular hallmarks of LSECs, suggesting that additional extrinsic cues from the specific liver environment are also required [107].

One overall limitation of most current studies characterizing endothelial cell heterogeneity is that the analysis only takes a snapshot of the endothelial cell functional, molecular, and morphological passport. Thereby these do not account for the dynamic nature of endothelial cells, which can quickly alter their phenotype in response to their changing environment. Time-lapse technologies can be part of the solution. In the case of the liver, it has been shown that the fenestrae are very dynamic structures and that their size, (re)appearance, and position can be monitored on live cells by four-dimensional time-lapse atomic force microscopy [114].

## 4. Organ-Specific Endothelial Cell Culture and Phenotypic Drift

During the last decades, an important advance in the characterization of organ-specific endothelial transcriptional and functional signatures has revolutionized the field. Despite this, proficient use of organ-specific endothelial cells in vitro is still conditioned to the difficulties for maintaining those features in culture [115]. One of the main challenges in tissue engineering and regenerative medicine, for example, is to obtain functional and vascularized (or vascularizable) grafts and organoids suitable for implantation or as models for in vitro studies. Several vascularized organoids are being developed so far by combining parenchymal cells of specific organs and endothelial cells in culture [116]. Besides, cultures of endothelial cells in monolayers represent a critical tool in mechanobiology, especially for traction force analyses [117]. In all cases, researchers must decide what source of endothelial is the most suitable. As can be observed in Table 1, many of them use commercially available organ-specific endothelial cells [118,119,120,121,122,123,124,125,126,127,128] and only a few freshly isolate the endothelial cells they work with [106,129,130,131,132]. Others use human embryonic or pluripotent stem cells (hESCs or hPSCs), co-/differentiated into endothelial cells in the presence (or not) of other stromal cells (such as pericytes or smooth muscle cells) or organotypic parenchymal cells [133,134,135]. Another option is to genetically modify organ-specific endothelial cells after isolation to force them to maintain their phenotypical cues for the longer term [124,125,136]. Yet, many researchers still chose insufficiently specialized endothelial cells, such as human umbilical vein endothelial cells (HUVECs) [137,138,139,140].

These approaches all face two common limitations. First, the characterization of specialized endothelial cells is still emerging and, thus, incomplete. Naturally, endothelial cells in 2D cultures or in 3D/organoids are being routinely characterized to validate these as mature organ-specific endothelial cells. Nevertheless, and specifically pointing at the hESC-derived endothelial cells, these mostly resemble the ones in embryonic developmental stages [134]. In this regard, Pappalardo et al. specifically observed how the maturity of human dermal blood endothelial cells stood out in the maintenance of vascularized skin grafts in the long term, compared with hPSC-derived endothelial cells and HUVECs [141]. The best way to determine the degree of validation for these endothelial cells would be to have the full genetic and phenotypic description of what each organ-specific endothelial cell should be like, but this knowledge is still limited. 

Alternatively, freshly isolated cells can be used as a control. Marcu et al. made a comparison between gene expression in human fetal heart, liver, lung, and kidney endothelial cells either freshly isolated or kept in culture over 2–5 passages. They identified that most of the differentially expressed genes between the two conditions were likely linked to the tissue microenvironment because these were not observed after in vitro culture, which decreased cell specialization [106]. For example, between heart and kidney, only 867 genes were differentially expressed after in vitro culture, whereas >5000 differentially expressed genes were identified in freshly isolated cells. This brings us to the second common limitation: after few passages, organ-specific endothelial cells lose many of their specific features if they do not meet their (not yet fully described) microenvironmental requirements. This phenomenon is known as phenotypic drift [9,142]. 

In an attempt to neutralize this loss, some groups have investigated the use of genetic modifications like an adenoviral genetic modification (transfection of E4ORF1) in primary endothelial cells (from lung and heart) to avoid cellular transformation along passages, preserving endothelial features in a longer-term [124,125]. Similarly, Palikuqi et al. transiently overexpressed the embryonic ETV2 factor in mature human endothelial cells in vitro (detailed in Table 1) and co-cultured these with organoid models (e.g., pancreatic islets), obtaining organ-specific endothelial cell cues [136]. However, the authors warn that their results may differ from those obtained by native endothelial cell counterparts [124,125]. The use of low specialized endothelial cells has become, therefore, one of the most suitable solutions to ensure a greater reproducibility in vascular research, even if such results have compromised specificity. Because of this, many studies are still based on the use of HUVECs (or other low specialized endothelial cells) as these are the best characterized endothelial cell types. Readouts from those are, regardless of the great heterogeneity in the endothelial cell population and its huge implications, generic. In brief, there is a profound need for reaching a comprehensive understanding of what features make organ-specific endothelial cells unique, and which of them can be rescued or maintained in culture through mimicking their microenvironmental cues. 

## 5. Role of the Tissue Microenvironment in Adult Endothelial Cell Heterogeneity

The improved resolution in “-omics” by single-cell and single-nuclei screening technologies is providing a constantly expanding picture of the heterogeneity within the endothelial cell population, as reviewed in Section 3. Their phenotype repertoire is strongly linked to the high diversity of microenvironments to which these cells are exposed. Thus, besides describing their genetic and functional signatures, the different contexts that these specialized cells are experiencing the need to be considered to picture out what are the extrinsic factors modulating their unique phenotypes. Within their respective tissues, endothelial cells are exposed to mechanical forces, such as shear stress and circumferential stretch by the blood flow, tissue stiffness, or pulsatility, and to biochemical factors, such as basement membrane or (more broadly) extracellular matrix (ECM) composition, cell-cell interaction, paracrine signaling pathways [143], cytokines [144], and other metabolites. Only by understanding the factors involved in conditioning their phenotypes can we begin to learn how to maintain and use these cells in culture.

### 5.1. Mechanical Cues Determining Endothelial Cell Heterogeneity

The different mechanical cues with effects on endothelial cell behavior or morphology have been recently reviewed elsewhere [145], but little emphasis is placed on the organ-specific effects. Here, we compile and discuss what has been reported on the mechanobiology of the endothelial cells’ heterogeneity. 

#### Organ-Specific Responses to Tissue Stiffness, Shear Stress, and Cellular Stretch

We know that endothelial cells can sense and react according to the mechanical stimuli that they experience in specific tissues. Yet, what are the phenotype changes associated with a given mechanical stimulus? Or, in other words, what are the stimuli in specific tissues and organs that make endothelial cells specialize and become heterogeneous? With the rise of mechanobiology [146], new tools have been developed to mechanically characterize living tissues from the macroscale to a nanoscale range [147]. The tissue stiffnesses for the brain, liver, and heart, for example, have been reported to be 0.6–2.4 kPa [148], 6 kPa [149], and 25 kPa on average [150], respectively. Thus, these measurements have facilitated the identification and comparison between different tissue (and cellular) stiffnesses under physiology and disease, as reviewed by Guimarães and colleagues [147]. For instance, tissue-specific stiffness was firstly reported to have an impact on cell lineage fate in 2006, when differently cross-linked polyacrylamide gels mimicking brain (0.1–1 kPa), muscle (8–17 kPa), and stiff matrices (25–40 kPa) proved to be neurogenic, myogenic or osteogenic, respectively, for naive mesenchymal stem cells [151]. Thereby, physiologically relevant tissue stiffnesses, as well as shear stresses, can nowadays be mimicked and applied in vitro to endothelial cells [128], composing more and more sophisticated models for mechanistic mechanotransduction and heterogeneity studies. 

Endothelial cells are affected by changes in their microenvironment stiffness with functional implications that can lead to disease. For instance, LSECs lose their fenestrations and become fibrotic when cultured on stiff matrices [149], or fuse and enlarge their fenestrations when conditioned to increased shear stress levels [152]. This is known to be associated with an increase in the production of nitric oxide (NO) by LSECs under higher shear stress levels [153], favoring a quick adaptation to changes in perfusion without affecting functionality.

However, when livers are fibrotic (or stiffer), although their endothelial cells experience greater levels of shear stress due to the augmented tissue resistance, their NO synthase (NOS) levels do not increase as in healthy LSECs [154]. The rate of NO production in response to flow has been shown to be dependent on the ECM stiffness in (bovine) aortic endothelial cells [155]. For HUVECs and immortalized human microvascular endothelial cells, the effect of the subendothelial stiffness was also studied and compared [156]. Based on traction force microscopy assays and transcriptomics, both cell types responded with an increase in their traction forces on stiffer substrates. However, variations in stiffness did not have an important impact on the transcriptome of these cells, and only a few stiffness-dependent genes were differentially expressed (i.e., upregulated TGF-β2 in HUVECs on stiffer matrices) [156]. 

Endothelial cells from different organs can behave differently under the same shear conditions. Under a 4 dynes/cm^2^ laminar shear, mouse microvascular pulmonary endothelial cells increase their cell stiffness and align with the flow direction, whereas cardiac endothelial cells do not, unless arachidonic acid is added [124]. When exposed to softer matrices (500 kPa polydimethylsiloxane or PDMS) instead of the classic polystyrene (PS) slides (2–3 GPa), under flow (2 dynes/cm^2^), cardiac endothelial cells showed more alignment on the stiffest substrate, whereas lung endothelial cells aligned and elongated more on the softer one [125]. However, although 500 kPa is more close to physiological stiffnesses than 2 GPa, limited tissues reach such levels in healthy conditions (e.g., nerves, cartilage, ligaments, tendons, bones [147], and heart valve leaflets [157]). In those studies, moreover, cardiac and lung endothelial cells were transfected with the E4ORF1 of the AdE4 gene complex to promote long-term survival as differentiated endothelial cells. Thus, it is unclear whether their native microvascular counterparts would have behaved equally. 

Under shear stresses up to 16 dynes/cm^2^, brain microvascular endothelial cells neither aligned with the direction of flow, contrasting with HUVECs, that did so [120]. Reinitz et al. point out that this may be a BBB pre-programmed feature that would not be affected by shear stress. Yet, these cells were cultured directly on PS surfaces (2–3 GPa), very far from the brain native stiffness (1–3 kPa, [147]). It is known that mesenchymal stem cells grown on soft matrices elongate to a greater extent and form stronger junctions in response to flow than those grown on stiffer matrices [158]. However, how the soft nature of the brain may affect the specialization of brain endothelial cells is still poorly understood [159]. Studies on brain endothelial cells are mostly focused on the BBB permeability, drug delivery, and the development of in vitro BBB models [160]. Shear stress in the brain increases the expression of tight junction proteins as well as their tightness [118,161], but only a few have characterized the impact that stiffness may have in such models [162]. There is likely little success in them forming monolayers on soft matrices, as evidenced by the improvement in monolayer coverage after stiffening collagen I gels with genipin cross-linkage [135]. In addition, tissue stiffness has implications on the glycocalyx composition of brain endothelial cells. A drastic reduction in the expression of heparan sulfate and Glypican 1 in the glycocalyx of brain-derived endothelial cells was reported after cultures on glass or plastic, compared to 2.5 or 10 kPa hydrogels [163]. 

Mechanical cues might facilitate the maintenance of structural heterogeneity of endothelial cells in culture. The formation of vascular networks was compared between fetal freshly isolated heart, liver, lung, and kidney endothelial cells embedded in collagen microfluidic channels [106]. After five days of gravity-driven flow, heart and lung cells form focal contacts, whereas kidney and liver cells form abundant fenestrations, being the liver ones discontinuous and irregular. Nevertheless, the extended signature lifespan from fetal endothelial cells might not be representative for adult organs [133,134].

Though heterogeneous reactions to shear stress have to some extent been studied, it is quite likely that stretch also plays a role, although it is studied to a much lesser degree. During inhalation, the alveolar–capillary barrier in alveoli stretches, and the ECM is compressed. Thus, besides shear stress and tissue stiffness, and as recently reviewed by Novak et al., pulmonary endothelial cells are markedly modulated by cyclic stretch during breath [164]. For instance, high amplitude cyclic stretch strain levels (i.e., 18 % elongation) impair pulmonary artery endothelial cell barrier function, while barrier integrity is protected at physiological levels (approx. 5%), with myosin light chain phosphorylation—thus, cytoskeleton rearrangement—being involved [121]. 

Importantly, cyclic stretch may also have a role in inflammation. High strains, such as 20%, have been shown to activate the production of pro-inflammatory cytokines in pulmonary endothelial cells, including IL-8, IL-6 and MCP-1 [165]. Additionally, calcium (Ca^2+^) influx in pulmonary endothelial cells is conditioned by cyclic stretch, being these channels activated at 20 to 30% cyclic stretch strains, but not at 10%, and mediated by actin polymerization [123]. Cyclic stretch and shear stress transduced by mitochondria has been recently suggested as a novel mechanotransducer having a role in Ca^2+^ influx in lung endothelial cells. Yamamoto et al. observed how shear stresses between 1 and 8 dynes/cm^2^ activated the mitochondrial production of adenosine triphosphate (ATP) and the Ca^2+^ influx mediated by Caveolin-1 [122].

### 5.2. Biochemical Cues Determining Endothelial Cell Heterogeneity

Most studies use generic protein coatings for their endothelial cell cultures, such as gelatin, fibronectin, or collagens, as shown in Table 1. However, the composition of the basement membrane and of the ECM in general, differs between organs and tissues, which may impact endothelial cell specialization, especially in the microvasculature [166]. Merna et al. described that lung and cardiac decellularized ECMs share several Collagens (I, III, IV, V, VI), Elastin, Fibrillin-1, Fibronectin 1 and Laminin, while Collagen II and IX were only present in lung, and Collagen VII, Fibrinogen and heparan sulfate proteoglycans were specific for cardiac ECMs [167]. Endothelial cells adhere to the basement membrane components through a repertoire of integrins. These are heterodimeric transmembrane receptors (composed of α and β subunits) that, moreover, are responsible for shear-induced mechanotransduction signaling through the reorganization of the cytoskeleton, among others [168]. Therefore, organ-specific ECM compositions are closely linked to endothelial cell phenotypes. 

In fact, using Fibronectin or Collagen alters the mechanosensitivity of endothelial cells [169] and of vascular smooth muscle cells [170]. Specifically, (bovine) aortic endothelial cells cultured on collagen and under shear stress showed only a transient (<30 min) activation of the small GTPase RhoA, responsible for cell stiffening–associated with pathological conditions; this was inhibited by the shear-induced PECAM1/integrin-dependent activation of PKA. The authors suggest that the abundant collagen found in the descending aorta may be atheroprotective. On Fibronectin, instead, shear stress did not activate PKA, and endothelial cells had a prolonged RhoA activation [169]. Thus, acknowledging the specific basement membrane components and taking them into account in cell cultures, together with mechanical cues, is key to reaching more physiologically relevant experiments. 

Bottom-up (or “discovery”) proteomics by liquid chromatography and tandem mass spectrometry on decellularized tissues and in vitro produced ECMs, are providing new insights in the characterization of the basement membranes and ECMs from different organs [171]. Different components of ECMs associated with specific organs have been reported so far [167,172], and a Matrisome database (MatrisomeDB), launched by Naba and Gao [173,174], can be accessed online [175]. Yet, the database is only recent and mostly contains proteomic data of bulk ECMs from some organs, including the human liver and colon, as well as some diseased tissues. Only a few basement membrane characterizations, such as for human retinal vascular and human glomerular basement membranes, are included; even if some large veins and arteries have been characterized so far, basement membrane data on organ-specific microvasculatures is not yet available. 

Interestingly, the aim of obtaining more physiologically relevant specific ECMs has led some groups to develop organ-specific ECM bio-inks from decellularized (porcine) tissues [176]. A less specialized commercial coating option offered by Corning is Matrigel, a mix of different ECM proteins produced by Engelbreth–Holm–Swarm mouse sarcoma tumor cells (including Laminin, Collagen IV, heparan sulfate proteoglycans, Entactin/Nidogen, and several growth factors). However, a common limitation for both decellularized ECM bio-inks and Matrigel is that they are batch-to-batch dependent, and thus they have an inherent variability associated that has not yet been traced. 

#### Organ-Specific Responses to Specific Biochemical Cues

In the absence of the complete composition data for basement membranes from organ-specific microvascular endothelial cells, the current best options to mimic their ECMs are: i) the use of co-cultures (either in 2D or in 3D) with cells that would produce the ECM in situ, or ii) the use of organ- or cell-derived ECMs (produced by the specific cell types of interest), regardless of their batch-to-batch limitation. Extracting the basement membrane from tissues may be cumbersome given the thin nature of this layer and the insolubility of many of its (mostly cross-linked) components [177]. Instead, Zhang et al. extracted the ECMs from skin, liver, and skeletal muscle decellularized tissues, and used them on (static) cultures with their respective endothelial cell types. They observed that when cells matched their ECM compositions, these showed a greater proliferation and differentiation capability [178]. Similarly, Bacci et al. produced and applied in vitro lung and cardiac ECMs to organ-specific endothelial cells, subjecting them to shear stress. Importantly, cardiac endothelial cells did not align under Fibronectin-coated surfaces [124,125] and aligned more on cardiac- and on lung-derived ECMs [125]. These studies demonstrate that when endothelial cells are cultured on generic coatings, part of their potential will be missing in vitro. 

Romero Liguoni et al. also produced three hydrogels from decellularized (porcine) left ventricle, mitral valve, and aorta ECMs [132]. They characterized them biomechanically and tested them for vascular matrix formation with pulmonary microvascular endothelial cells co-cultured with adipose stromal cells. Collagen VI was the most abundant protein in the left ventricle and mitral valve matrices, while Elastin was the most abundant in aorta ECM. All three supported vascularization, but the ventricle ECM showed the greatest structures. Of note, aortic ECM was the stiffest one (around 7 kPa) and supported adipose cell myogenic differentiation, whereas valve and ventricle ECMs were both 3 kPa and inhibited an induced (TGF-β1) myogenic differentiation [132]. 

Decellularized kidney and liver ECMs have also been successfully used to maintain in vitro the viability and proliferation of human glomerular endothelial cells [179] and vascularized liver organoids [138], respectively. In the liver case, however, whole livers were decellularized and used as scaffolds. With regard to the brain, ECMs have been extracted and their compositions have been compared with the native tissue and Matrigel [180] or between different brain regions [181] using mass spectrometry; however, no endothelial studies have been performed with those yet. Recently, the use of organ-specific decellularized ECM hydrogels [182] and decellularized ECM scaffolds [183] have been reviewed elsewhere.

Endothelial cells are also exposed to tissue-specific soluble components in their native microenvironments. As commented in the previous section, the addition of arachidonic acid in the media was required for cardiac endothelial cells to align with the flow at 4 dynes/cm^2^ [124]. Moreover, its inhibition in pulmonary endothelial cultures reduced their alignment to flow and their cell stiffness [124]. In another study, under the same conditions, heart endothelial cells had the greatest angiogenic capability after VEGF supplementation (also greatest oxygen consumption and glycolysis rates) compared to kidney, lung, and liver endothelial cells [106]. 

Although these have usually received little attention, paracrine signaling factors and pathways also have an organ-specific important impact on endothelial cells. Noteworthy, as recently exposed by Ricard et al., genetic variations in some signaling factors may lead to vascular diseases affecting specific organs instead of the vasculature in all organs [143]. That is, hereditary hemorrhagic telangiectasia (or HHT) caused by mutations in *ACVRL1* and *ENG*, involves arteriovenous malformations mainly affecting the lungs and liver; or, mutations in *BMPR2* may lead to pulmonary hypertension with no impairment of the endothelium in other organs [143].

Oxygen, or its absence during hypoxia, is known to regulate endothelial cell homeostasis. Hypoxia-inducible factor (HIF)-α subunits are hydroxylated and degraded under normal oxygen tension conditions. During hypoxia, apoptosis, oxidative stress, and angiogenesis are increased in an oxygen-dose and time-dependent manner: during short and mild exposures, anti-apoptotic pathways (i.e., NFκB signaling and HIF1α and γH2AX DNA repair pathways) are activated. Instead, severe or longer exposures to mild hypoxia can lead to an increase in HIF1α and promote apoptotic pathways [184]. The rise of HIF transcription factors in HUVECs after hypoxia has been suggested to increase vascular permeability, triggering inflammatory effects driven by the CD34-mediated differentiation of endothelial cells [185]. In the brain, chronic mild hypoxia disrupts the permeability in the BBB. Brain regions with greater hypoxia-induced vascular leakages also show the greatest angiogenic remodeling [186]. Thus, the increase in permeability is likely linked to a compensatory vascular remodeling (proliferation and angiogenesis), consistent with the switch of the normally expressed integrin α6β1 by α5β1 after cerebral hypoxia [187]. 

The adaptation of the endothelial cells to different levels of hypoxia occurs through a two-step mechanism involving initiation of angiogenesis (lead by HIF-1) and maturation of the vascular network (lead by HIF-2) [188]. Bartoszewski et al. reported that this HIF-1 to HIF-2 transition during hypoxia is conserved among nine bed-specific endothelial cell types (i.e., microvascular cardiac, dermal, lung, and uterine; aortic, cardiac artery and iliac; pulmonary artery endothelial cells, and HUVECs) [127]. However, HIFα subunits do have bed-specific behaviors under hypoxia modulating the ECM deposition by endothelial cells. Collagen type IV and Fibronectin are secreted by human umbilical artery endothelial cells (HUAECs) and HUVECs under both physiological (5% O_2_) and hypoxic (1% O_2_) conditions, whereas collagen type I is only deposited under physiological O_2_ levels [140]. Remarkably, the production of Collagen type IV and Fibronectin by HUAECs is mediated by HIF1α, whereas HIF2α drives this process in HUVECs, suggesting tissue-specific mechanisms in different endothelial cell types. In addition, more immature cells (endothelial colony-forming cells) secrete collagen type I under physiological O_2_ conditions, but not Fibronectin and Collagen IV, which are deposited under saturated levels of O_2_ (or normoxia) [140].

## 6. Impact of Endothelial Cell Heterogeneity on Drug Development

Given the high diversity of phenotypes within the endothelial cell population and given their differential behavior in response to extrinsic factors, it is not surprising that the use of generic targets in vascular repair, revascularization, or in cancer treatments (such as VEGFR) entail low efficiencies and suffer from side effects [189,190]. New molecular and genetic knowledge in endothelial heterogeneity implies that endothelial cells from different origins may not respond equally to drugs. In fact, organ-specific endothelial cells show differential reactivity to inflammatory stimuli, such as cytokines, and this entails pharmacological implications [144]. In this regard, more research is being focused on the characterization of the heterogeneous inflammatory response of endothelial cells for the development of anti-inflammatory therapies against chronic inflammatory diseases [191]. 

In recent decades, the unique properties of some endothelia have been studied and exploited in the development of therapeutics. In cancer, the heterogeneity of tumor endothelial cells has proven to present interesting opportunities for drug delivery, as has been reviewed elsewhere [192]. Interestingly, the unique properties of the tumor vasculature have been demonstrated to carry a better potential for tumor targeting than the tumor cells themselves, and novel anti-tumor strategies are emerging [193,194]. Another endothelium that has raised a great interest in drug development is that in the BBB. The tight nature of this vasculature becomes a challenge in terms of drug delivery and, thus, in vitro BBB models are being developed to test brain-targeting drugs [195,196]. Despite their debatable resemblance with their primary counterparts, patient-specific endothelial cells derived from pluripotent stem cells have emerged as a promising tool with the added value of their potential for more personalized drug development [197]. While the heterogeneity of endothelial cells represents a challenge for designing tailored revascularization strategies, it also generates opportunities for vascular bed-specific and more efficient delivery of drugs, as recently reviewed [198]. 

Still, the knowledge on endothelial heterogeneity across different organs, tissues, and species is constantly increasing, and a giant leap towards its full integration and consideration in the design of new drug delivery strategies, as well as in tissue engineering and regenerative medicine, must be taken to reduce off-target effects. 

**Table 1 ijms-23-01477-t001:** Organs targeted for the characterization of endothelial cells using in vitro culture techniques, including the strategies used for the maintenance of organ-specific endothelial cell signatures.

Organ/ Tissue	Cell Type	Pass. Nr.	Co-Culture	Tissue Mimicking		Characterization Technique			Time in Culture	Refs
				Mechanical	Biochemical	Genetics	Morphology	Function		
Brain	human brain microvascular (mv) ECs (HBMECs) (C)	P2–P3	human astrocytes	6.2 dynes/cm^2^; PP hollow fibers	FN	RNA microarray	-	TEER; glucose consumption and lactate production	30 d	[118]
		P4–P7	-	10–20, 40 dynes/cm^2^; silicone	FN + Astrocyte conditioned medium	-	IF: CD31, ZO-1 and CLDN-5; WB: Transport markers P-gp and GLUT1	Src/ERK pathway activation	4 d	[161]
	bovine primary BMECs (F)	P1–P7	Glial cells (astrocytes >95%) (F)	-	Col solution	-	IF: CLDN, OCLN, ZO, β-cat, p120^cat^, actin cytoskeleton	Permeability assays	14 d	[129]
	mouse primary BMECs (F)	P1	-	-	Matrigel	RNA-seq, transcriptome	IF: CLDN-5, OCLN, ZO-1, ZO-2, JAM-A, VE-cad & β-cat	TEER	7 d	[130]
	mouse primary BECs (F)	P1	-	-	Col I	RNA-seq and ATAC-seq	IF: CD31	-	10 d	[131]
	iPSCs-derived HBMECs & human umbilical vein ECs (HUVECs) (D)	P1–P7	-	~4 dynes/cm^2^; cylindrical 150 μm Ø channel Col hydrogel	Col I	GLUT1 and P-gp expression	IF: ZO-1, CLDN-5 and OCLN	Permeability assays	6 d	[133]
	iPSCs-derived HBMECs (D)	P2	-	-	Genipin-crosslinked Col I gels, with FN and Col IV	-	IF: ZO-1 and CLDN-5	TEER, microvessel formation	7 d	[135]
	hESCs-derived ECs (D)	-	hESCs-derived cortical organoids	Perfusion tests	cortical organoids	TJ & nutrient transporter expression; single-cell map vhCOs	-	TEER	120 d	[134]
	Immortalized mouse BMECs (bEnd3) (C)	-	patient-derived glioblastoma cells	Alginate fibers	thiolated sodium hyaluronate	qPCR	IF	VEGF release	14 d	[119]
	Immortalized HBMECs; HUVECs (C)	-	-	8, 12, 16 dynes/cm^2^	-	-	IF: F-actin and ZO-1; WB: β-catenin and ZO-1; cell alignment	-	36 h	[120]
Lung	human pulmonary artery ECs (HPAECs) (C)	P6–P8	-	flexible-bottomed BioFlex plates; 5 and 18% elongation cyclic stretch	Col I	gene profiling	IF: F-actin; stress fiber & actin alignment; WB: pathway factors	cytoskeletal rearrang. & TEER	2 d	[121]
		P7–P10	-	1, 3, 8 dynes/cm^2^; glass	Gelatin	-	IF: MitoTracker and caveolin-1	Real-time imaging: mit. ATP levels; Ca^2+^ influx	few min	[122]
	human pulmonary mv ECs (HPMECs) (C)	P4–P7	-	silicon chamber; 10, 20, 30% stretch strains	FN	qPCR: TRPV-2, TRPV-4	IF: Tie-2, CD31, F-actin	Stretch-activated Ca^2+^ influx	few min	[123]
	mouse primary PMECs & cardiac mv ECs (both E4ORF1) (C)	-	-	4 dynes/cm^2^; PS slides	FN	-	FC: CD31, CD144; cell alignment & area; AFM: cell stiff.	-	12 h	[124]
		-	-	2 dynes/cm^2^; PDMS (500 kPa) and PS (2–3 GPa) slides	Cardiac & lung ECM vs. FN	-	cell alignment and area; FC: integrins αv and β3	-	12 h	[125]
Heart	bovine primary aortic ECs (F)	-	-	12 dynes/cm^2^; glass; 100 pN pulsatile & 10 pN continuous forces	FN or Col I	-	WB: RhoA, ph-CREB, ph-PKA, PKA, ph-serine; IF: actin, vinculin, β-cat, HUTS-4, VE-cad	Bead displacement by pulsatile force; cAMP; integrin activation	30 min	[169]
	HPMEC-ST1.6R (F)	-	Adipose tissue-derived stromal cells	Left ventricle-, mitral valve-, aorta-derived hydrogels (3, 3, 7 kPa)	Left ventricle, mitral valve & aorta ECM	-	IF: SM22α, actin, CD31	Vascular network formation	7 d	[132]
Liver	HUVECs (F)	-	fetal liver cells	perfusion at 0.5 mL/min; liver decellularized scaffolds	Liver decellular. ECMs; matrigel	-	IF: vWF, eNOS, Ki67, TUNEL	Vascular network formation; prolif.; platelet deposition	7 d	[138]
Unspecific	HUVECs (C)	-	-	20 dynes/cm^2^; ibidi slides	-	qPCR: Wnt ligands	cell polarity & orientation; IF: Cleaved Caspase-3, Col IV, Erg1/2/3, GM130, Golph4, ICAM2, Lef1, NG2; FC: CD31, CD45	-	4 h	[139]
	HUVECs (C)	P6–P10	THP1 cells	FITC-conjugated dextran flow	15(S)-hydroxyeicosatetraenoic acid	-	IF: ZO-1, OCLN	Barrier permeability & disruption; THP1 transmigration	8 h	[199]
	bovine aorta ECs & HUVECs (C)	P6–P10	-	6, 12, 18 & 22 dynes/cm^2^; 100 Pa, 2.5, 3, 10 & 30 kPa PAA gels	FN	-	cell alignment & area; IF: actin, NF-κB	TNF-α induced NF-κB transloc. to nucleus	24 h	[128]
	human pulmonary artery ECs (HPAECs) (C)	P5–P9	-	1.1 & 40 kPa hydrogels, or glass (~50 GPa)	FN or Col IV	-	IF: VE-cad, paxillin, actin	Magnetic twisting cytometry for VE-cad receptor perturbation & displacement; Monolayer stress microscopy	5 d	[200]
	immortalized human mv ECs (HMEC-1) & HUVECs (C)	P4–P8	-	3, 35 & 70 kPa PAA gels	Col I	Transcriptom. and qPCR	IF: pMLC & actin & WB	Traction force microscopy	2 h	[156]
	human umbilical artery ECs (HUAECs) & HUVECs (C)	-	-	-	Col I; hypoxia	qPCR: β-actin, HPRT1	FC: VE-cad, CD31, KDR, CD146, PDGFRβ; IF & WB: Col I, Col IV, FN, laminin, actin	Hypoxia & conditioned ECM deposition	7 d	[140]
Diverse	fetal human primary kidney, lung, liver & heart ECs (F)	P2–P5	rat primary hepatocytes	Gravity-driven flow; cells in Col microfluidic channels	Col I	RNAseq of freshly isolated vs. cultured ECs	IF: CD31, CD144, vWF, PV1 & Caveolin 1	TEER, spheroid sprouting, metabolic assays	5 d	[106]
	human primary mv dermal, lung, renal glomerular, brain & liver ECs; large vessel coronary artery ECs & HUVECs (C)	P2–P8	-	-	Dilutions of TTP/sporadic HUS patients’ plasma	qPCR: Fas transcripts	FC: annexin II	Apoptosis: Cdc2 kinase assay, procoagulant activities	16-18 h	[126]
	human primary mv cardiac, dermal, lung & uterine ECs; aortic, cardiac artery, iliac ECs, HPAECs & HUVECs (C)	P2–P6	-	-	Hypoxia	Gene expression microarray; qPCR: HIF1A, HIF-2a, 18S, TBP	WB: HIF-1a, HIF-2a; β-actin	Hypoxia effects in transcriptome	2 d	[127]
	human adipose-derived endothelial cells & HUVECs (F); human mv cardiac, aortic, pulmonary and dermal ECs (C); ETV2-transduced.	P5–P10	Colorect. cancer, colon & small intestine organoids;pancreat. islets	Gravity-driven perfusion tests in microfluidic devices	Matrigel or mixture of laminin, entactin & col IV	single-cell transcriptom. & epigenetics	IF: VE-cad, CD31, PDGFRβ; FC: CD31, CD45; WB: RAP1, ETV2, ETS1, p-AKT; vessel area	Vascular tube formation; glucose-responsive insulin-secreting (islets); intestine & organoid vascularization	12 w	[136]

Pass., passage; C, commercial; D, derived; F, freshly isolated; ECM, extracellular matrix; Col, collagen; FN, fibronectin; FC, flow cytometry; IF, immunofluorescence; WB, western blot; PAA, polyacrylamide; PDMS, polydimethylsiloxane; PP, polypropylene; PS, polystyrene; ph, phospho-; AFM, atomic force microscopy; TEER, transendothelial electrical resistance; TJ, tight junction; β-cat, β-catenin; CLDN, claudin; CREB, cAMP-response element binding protein; ETV2, ETS variant transcription factor 2; HIF, hypoxia-inducible factors; ICAM, Intercellular Adhesion Molecule; JAM, junctional adhesion molecules; KDR, Kinase insert domain receptor; NG2, Neuron-glial antigen 2; OCLN, occludin; PDGFRβ, Platelet-derived growth factor receptor β; PV1, plasmalemma vesicle-associated protein; RAP1, Ras-Association Proximate 1; SM22α, Smooth muscle protein 22α; VE-cad, VE-cadherin; vWF, Von Willebrand factor; ZO, zona occludens; d, days; h, hours; w, weeks; (-), not mentioned data.

## 7. Conclusions

In recent decades, both mechanical and biochemical cues have been demonstrated to have a huge impact on the behavior of organ-specific endothelial cells. Nevertheless, in most in vitro research, these cells are being cultured on plastic or glass surfaces, under static conditions, and/or on generic protein (mostly single) coatings to promote cell attachment. Furthermore, difficulties in keeping tissue-specific characteristics have led to the use of insufficiently specialized endothelial cell types in most research papers. New transcriptomic and proteomic insights are being revealed for the characterization of these bed-specific endothelial cells. Furthermore, new approaches are being developed to mimic the endothelial cell microenvironment, such as hydrogels and microfluidic strategies. In addition, biophysicists and molecular biologists are joining forces to improve the current understanding of organ-specific vascular mechanobiology. Therefore, it is crucial to put all this knowledge and technological potential together to facilitate the transition from the generic characterization of the endothelial cell behavior to more specialized readouts. 

## Figures and Tables

**Figure 1 ijms-23-01477-f001:**
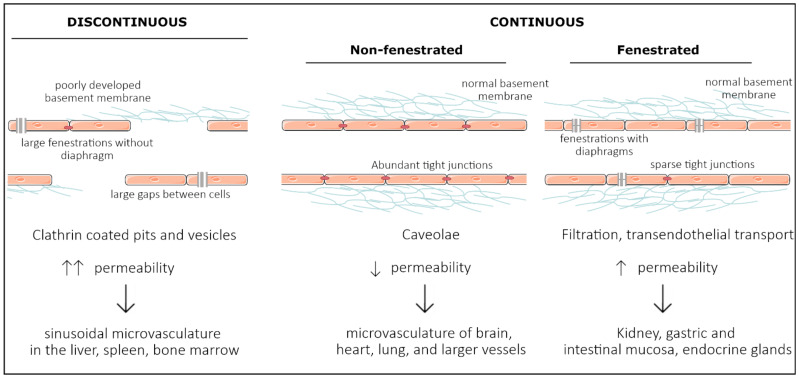
Representation of the three main structural phenotypes in organ-specific microvasculature. Discontinuous endothelium is mainly found in the sinusoidal microvasculature of the liver, spleen and in bone marrow, and is characterized by large fenestrations and pores within and in between endothelial cells, respectively. It has a poorly developed basement membrane and contains clathrin-coated pits and vesicles that dramatically increase permeability. Non-fenestrated endothelium is characterized by low permeability and a high abundance of tight junctions and caveolae. It is mostly found in the microvasculature of the brain, heart, and lung and in larger vessels. Fenestrated endothelium has an intermediate permeability and is characterized by fenestrations covered with a diaphragm. These fenestrations and sparse tight junctions ensure proper filtration and transendothelial transport, as found in the microvasculature of kidney, gastric and intestinal mucosa, and endocrine glands.

**Figure 3 ijms-23-01477-f003:**
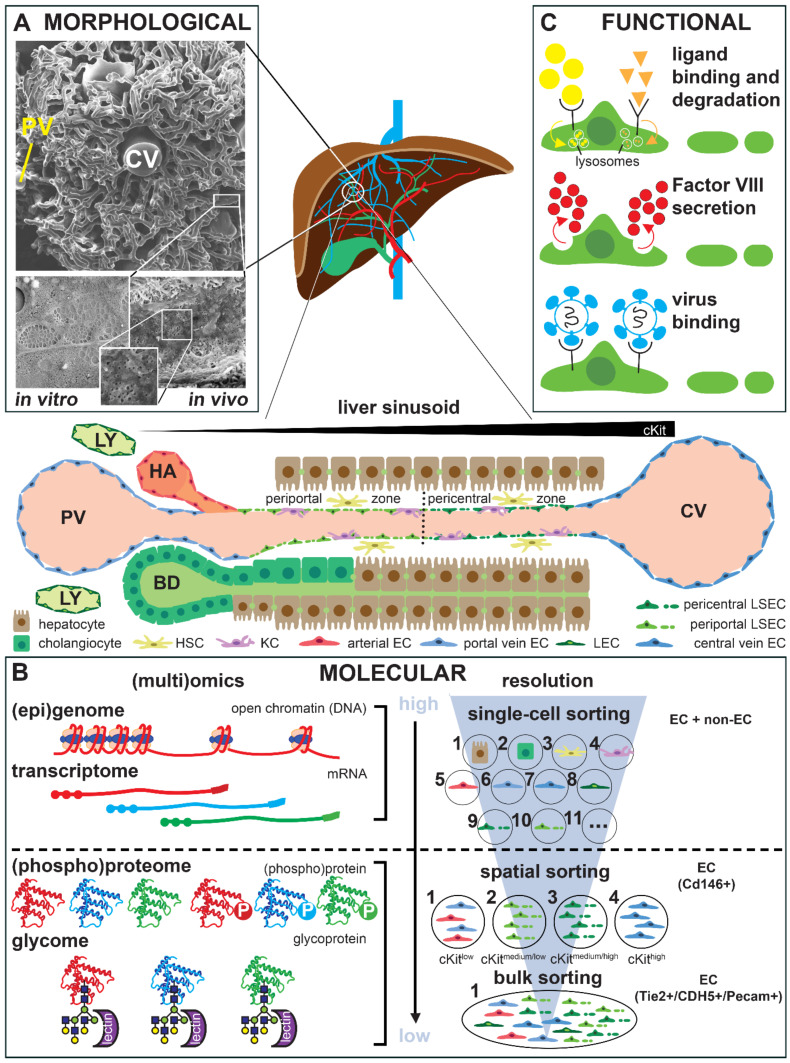
Morphological, molecular, and functional characterization of liver endothelial cells. Central panel. Schematic representation of a liver sinusoid with the different cell types and the zonated organization of the liver sinusoidal endothelium. Note the increasing gradient of cKit expression from the portal nodules to the central venule. HA: hepatic arteriole; BD: bile duct; PV: portal venule; CV: central venule; LY: lymphatic capillaries. (**A**) Morphological characterization of the liver lobular vasculature (*top*), liver sinusoids (bottom, right) and freshly isolated liver sinusoidal endothelial cells (LSECs; bottom, left) by scanning electron microscopy revealing fenestrae organized in sieve plates. (**B**) Molecular characterization of liver endothelial cells at different omics levels (left) and at different cellular resolutions (right). The dotted line indicates that single-cell resolution is currently routinely possible for the epigenome and the transcriptome, but not (yet) for the (phospho)proteome or glycome. The glycome can be analyzed indirectly by specific lectin binding patterns. P: phosphate. (**C**) Functional characterization of LSECs showing three commonly used assays, i.e., ligand binding and lysosomal degradation (top), Factor VIII secretion (middle), and virus binding (bottom).

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
