# Peer review of "Organ-Specific Endothelial Cell Differentiation and Impact of Microenvironmental Cues on Endothelial Heterogeneity"

_ijms, 2022, doi:10.3390/ijms23031477_

Round 1

Reviewer 1 Report

The heterogeneity of endothelial cells is highly correlated with the specific functions carried out by organs and tissues throughout the body.  The degree of heterogeneity in differentiation is already determined from the first steps of development, including the specification of arterial and venous pathways to the determination of microvascular fates.  In order to develop more specialized tissue engineering and vascular bed repair techniques, it is critical to recognize the differences in phenotypes of endothelial cells.    Even though novel transcriptomic and proteomic technologies are facilitating the unravelling of vascular bed-specific endothelial cell signatures, there are still many studies that employ inadequately specialized endothelial cells in their investigations. Not only are endothelial cells heterogeneous, but their specialized phenotypes are also dynamic and react to changes in their microenvironment.    A growing body of work reveals how mechanical and biochemical context modulates endothelial cells by combining chemistry, molecular biology, and computational disciplines in recent decades. The lack of specialization of endothelial cells still prevents a full understanding of phenotypic differences between organs that are determined by tissue-specific biomechanical factors. Therefore, the authors focus this review article on how developing and adult endothelial cells' signatures are determined by their microenvironment.  In this article, the authors discuss the most recent research conducted on endothelial cells and the implications of mimicking tissue-specific biomechanical cues in culture.   In general, I found the manuscript very interesting to read. However, I have some minor comments to improve the manuscript insightful.     1. Author should comment on, ' species-specific endothelial heterogeneity' within the manuscript. 2. Author should dedicate a subtitle and discuss, how does endothelial heterogeneity impacts drug development.     

Author Response

We thank Reviewer 1 for his/her positive comments on the manuscript. The addition of the two suggested matters has made the manuscript more complete than in its previous form.

  1. Author should comment on, ' species-specific endothelial heterogeneity' within the manuscript.

Response 1: We briefly mentioned this issue in section 3 and have now expanded this with inclusion of the notion that this issue has implications for translation of animal findings to patients (page 6, section 3).

  1. Author should dedicate a subtitle and discuss, how does endothelial heterogeneity impacts drug development. 

Response 2: A small section (page 15, section 6) has been added at the end of the review addressing the impact of endothelial cell heterogeneity on the development of new drugs and treatments.

Reviewer 2 Report

In this review, Gifre-Renom et al presents the organ-specific endothelial cell differentiation. This review is well written and of high interest. I have minor comments to improve the manuscript.

  • Figures could be a little bit better.

On Figure 1, fenestrations are not well represented, as they are windows going through endothelial cells and not between endothelial cells. They should be represented differently on the right panel and are missing on the left panel (discontinuous endothelial have gaps between endothelial cells but also fenestrations within endothelial cells). I would also suggest mentioning that discontinuous endothelium is also found in bone marrow and spleen and fenestrated endothelium in endocrine glands.

On Figure 2, the capillary representation is with 2 layers on endothelial cells (on the right), where endothelium is a single layer organization. For the liver endothelium, in the column organ specific environment, it is written “GATA4 secreted by epithelial cells”. GATA4 is a transcription factor specifically expressed in liver sinusoidal endothelial cells and is not expressed or secreted by epithelial cells. I would suggest modifying the incorrect statement with BMP9 secreted by hepatic stellate cells (Desroches-Castan et al, hepatology, 2019) which showed that BMP9 secreted by hepatic stellate cells induced terminal differentiation of liver sinusoidal endothelial cells and upregulates GATA4 in these cells.

On Figure 3, panel B, the phosphoproteome illustration and the glycome illustration are confusing as they are fused. I would suggest separating them.

  • Line 42/43 I would add endocrine glands in the list of fenestrated endothelium.
  • Line 48/49, for the kidney, glomeruli endothelium is also fenestrated to ensure proper filtration, in addition to peritubular capillaries already mentioned.
  • On page 7, the authors mentioned Inverso et al (ref 92). I think the authors should describe a little bit more that work. Indeed, the authors chose to focus on the hepatic endothelium and this work combines RNAseq, phosphoproteomic and proteomic on hepatic endothelial cells to define zonation expressions of RNA, proteins and protein phosphorylation. It is to my knowledge pretty unique to spatially combine these 3 techniques. They also created a website to easily access the data (https://pproteomedb.dkfz.de/). I would suggest to mention this work a little bit more.
  • On page 9, section 4, I would add another work: Palikuqi et al, Nature 2020. In this paragraph, the authors present the challenge of keeping organ specific endothelial cells in culture. It is worth mentioning that Palikuqi et al transiently overexpressed ETV2 transcription factor in endothelial cell in vitro to create “naïve” endothelial cells and they could then differentiate these endothelial cells in an organ-specific manner in organoid models (notably pancreatic islets). I believe this paper should be cited in this paragraph.
  • Page 10, section 5, the authors present how microenvironment create endothelial cell heterogeneity. This part is very interesting but focus only on a subset of microenvironment with the mechanical forces and the integrin-ECM interaction. Paracrine signaling pathways are largely ignored. I would suggest citing a review on these signaling pathways (such as Ricard et al, Nat Reviews Cardiology, 2021) in the introduction paragraph of this 5th section to complete the section.

Author Response

We thank Reviewer 2 for his/her positive comments and helpful suggestions. We believe these have improved the content in the manuscript.

  • Figures could be a little bit better.

Response 1: All figures have been modified according to the Reviewer’s suggestions.

On Figure 1, fenestrations are not well represented, as they are windows going through endothelial cells and not between endothelial cells. They should be represented differently on the right panel and are missing on the left panel (discontinuous endothelial have gaps between endothelial cells but also fenestrations within endothelial cells). I would also suggest mentioning that discontinuous endothelium is also found in bone marrow and spleen and fenestrated endothelium in endocrine glands.

Response 2: The representations for the fenestrations have been modified in Figure 1, such that they are clearly within endothelial cells. Bone marrow and spleen have been added in the list for discontinuous endothelium and endocrine glands in the list for fenestrated endothelium, in Figure 1. These have also been mentioned in the text (pages 1-2, section 1).

On Figure 2, the capillary representation is with 2 layers on endothelial cells (on the right), where endothelium is a single layer organization.

Response 3: Thanks to this comment we realized that pericytes being pictured next to the endothelial cells in the capillary illustrations (capillary, brain, liver, and lung) were easily confused with a second layer of endothelial cells. We have now removed pericytes from the illustration since they were not giving any relevant information in Figure 2, and now only the endothelial monolayer is depicted.

For the liver endothelium, in the column organ specific environment, it is written “GATA4 secreted by epithelial cells”. GATA4 is a transcription factor specifically expressed in liver sinusoidal endothelial cells and is not expressed or secreted by epithelial cells. I would suggest modifying the incorrect statement with BMP9 secreted by hepatic stellate cells (Desroches-Castan et al, hepatology, 2019) which showed that BMP9 secreted by hepatic stellate cells induced terminal differentiation of liver sinusoidal endothelial cells and upregulates GATA4 in these cells.

Response 4: This statement has been now corrected in Figure 2 and included in the text (page 5, section 2.3).

On Figure 3, panel B, the phosphoproteome illustration and the glycome illustration are confusing as they are fused. I would suggest separating them.

Response 5: Panel B in Figure 3 has been modified according to the suggestion, so that the phosphoproteome and the glycome are separated.

  • Line 42/43 I would add endocrine glands in the list of fenestrated endothelium.

Response 6: The endocrine glands have been added in the text and in Figure 2 as fenestrated endothelium.

  • Line 48/49, for the kidney, glomeruli endothelium is also fenestrated to ensure proper filtration, in addition to peritubular capillaries already mentioned.

Response 7: The glomeruli endothelium has been added in the sentence as being also fenestrated, together with the peritubular capillaries.

  • On page 7, the authors mentioned Inverso et al (ref 92). I think the authors should describe a little bit more that work. Indeed, the authors chose to focus on the hepatic endothelium and this work combines RNAseq, phosphoproteomic and proteomic on hepatic endothelial cells to define zonation expressions of RNA, proteins and protein phosphorylation. It is to my knowledge pretty unique to spatially combine these 3 techniques. They also created a website to easily access the data (https://pproteomedb.dkfz.de/). I would suggest to mention this work a little bit more.

Response 8: The work from Inverso et al. has been further detailed in the text as suggested (page 7, section 3).

  • On page 9, section 4, I would add another work: Palikuqi et al, Nature 2020. In this paragraph, the authors present the challenge of keeping organ specific endothelial cells in culture. It is worth mentioning that Palikuqi et al transiently overexpressed ETV2 transcription factor in endothelial cell in vitro to create “naïve” endothelial cells and they could then differentiate these endothelial cells in an organ-specific manner in organoid models (notably pancreatic islets). I believe this paper should be cited in this paragraph.

Response 9: The work from Palikuqi et al., Nature 2020 has been included in the text (page 11, section 4) as suggested, as well as in Table 1.

  • Page 10, section 5, the authors present how microenvironment create endothelial cell heterogeneity. This part is very interesting but focus only on a subset of microenvironment with the mechanical forces and the integrin-ECM interaction. Paracrine signaling pathways are largely ignored. I would suggest citing a review on these signaling pathways (such as Ricard et al, Nat Reviews Cardiology, 2021) in the introduction paragraph of this 5th section to complete the section.

Response 10: We agree with Reviewer 2 that this information was missing, and the work from Ricard et al. has been now added in the introduction paragraph from section 5 (page 10), and a small paragraph related to it has been included in section 5.2.1 (page 15).
